# Distribution Patterns and Ecological Determinants of Suitable Habitats for the Dhole (*Cuon alpinus*) in China

**DOI:** 10.3390/ani15040463

**Published:** 2025-02-07

**Authors:** Yuangang Yang, Peng Luo, Yu Zhao, Tongzuo Zhang, Feng Jiang, Zhangqiang You

**Affiliations:** 1Ecological Security and Protection Key Laboratory of Sichuan Province, Mianyang Normal University, Mianyang 621000, China; yangyuangang92@foxmail.com (Y.Y.); q735680427@gmail.com (P.L.); 18748539460@163.com (Y.Z.); 2Qinghai Provincial Key Laboratory of Animal Ecological Genomics, Northwest Institute of Plateau Biology, Chinese Academy of Sciences, Xining 810008, China; zhangtz@nwipb.cas.cn; 3Key Laboratory of Adaptation and Evolution of Plateau Biota, Northwest Institute of Plateau Biology, Chinese Academy of Sciences, Xining 810008, China

**Keywords:** dhole, MaxEnt model, potential suitable habitat, environmental factors, wildlife conservation

## Abstract

This study aimed to identify the distribution patterns of potential suitable habitats for the dhole in China and elucidate the key ecological factors influencing their suitability. The analysis revealed that temperature changes, extreme low temperatures, and elevation significantly shaped the distribution of suitable habitats for dholes. Potentially suitable habitats were primarily concentrated in the central-western and northwestern regions of China. Additional scattered distributions were found in southeastern areas. Notably, Guizhou, Sichuan, and Guangxi Provinces exhibited the highest proportion of suitable habitat area. These findings provide essential ecological insights to inform conservation and management strategies for this species in China.

## 1. Introduction

Habitat is defined as the complex interplay of biotic and abiotic factors that are essential for the survival and reproduction of animals. These factors encompass food resources, water, and shelter, which are indispensable for the sustenance of animal populations [1]. Habitat suitability assessment is predicated on an analysis of the specific needs of animals within their habitats, coupled with an evaluation of the environmental factors that influence their survival and reproductive success. The goal is to delineate the areas within a certain range that are conducive to the survival of the target species [2]. Currently, global biodiversity is undergoing a relentless decline. The Red List of the International Union for Conservation of Nature (IUCN) indicates that over 4630 species are currently at risk of extinction including 26% of mammals, 41% of amphibians, and 12% of birds (https://www.iucnredlist.org/, accessed on 30 December 2024). The fundamental step in advancing global biodiversity conservation is the elucidation of the distribution patterns of endangered species and the identification of key environmental factors through habitat suitability assessments.

The ecological niche model (ENM) has emerged as a widely utilized approach for assessing habitat suitability for species [3]. By integrating diverse environmental variables, ENMs simulate and predict the potential spatial distribution of species, providing a quantitative and spatially explicit framework. This methodology facilitates an understanding of the interactions between landscape-level biological and environmental attributes and the ecological and geographical distribution of species [4,5]. Among the various ENMs, prominent examples include MaxEnt, Domain, ENFA, and GARP [6]. The MaxEnt model, in particular, has found extensive application in predicting and mapping the potential distribution of suitable habitats for a wide range of animal taxa including birds, ungulates, and carnivores [7,8,9]. For instance, the MaxEnt model has been employed to analyze the effects of different climate models on the potential distribution of suitable habitats for the snow leopard (*Panthera uncia*) [10].

The dhole (*Cuon alpinus*, Pallas, 1811), a member of the Canidae, Carnivora, is recognized as a top predator in forest ecosystems and plays a pivotal role in regulating the population structure and dynamics of these biomes [11]. As a highly social and diurnal canid, the dhole exhibits specific behavioral characteristics that distinguish it from other members of the Canidae [11]. Historically, the dhole was widely distributed across most Chinese provinces including Guizhou, Sichuan, Guangxi, and other regions [12]. However, during the mid-to-late 20th century, the dhole populations in China experienced a rapid decline due to a confluence of factors including habitat loss and fragmentation, poaching, and other historical issues [13,14]. Currently, the dhole in China is facing a significant survival crisis [15]. It has been upgraded to a Level I key protected animal in China (2021) and is classified as an “endangered” species by the IUCN [16]. Research on the dhole in China has focused on several critical areas including phylogeny and evolution [17], distribution range and quantity [12], captive population feeding and treatment of diseases [18], and the composition and diversity of gut microbes [19]. Nevertheless, the distribution pattern of potential suitable habitats and its key influencing ecological factors for the dhole in China remain unknown.

Here, we selected the dhole in China as the focal species and utilized the MaxEnt model to address two key objectives: (*i*) identifying the primary environmental factors influencing the potential suitable habitat for the dhole in China, and (*ii*) delineating the spatial distribution patterns of this potential suitable habitat. Through a comprehensive assessment of habitat suitability, this study aimed to provide critical ecological data to inform and support conservation and management strategies for the dhole across its range within China.

## 2. Materials and Methods

### 2.1. Collection and Organization of Dhole Distribution Site Data in China

The MaxEnt model, developed by Steven Phillips and colleagues based on the principle of maximum entropy, is a widely used ecological niche model (ENM) for density estimation and species distribution prediction [20]. It is primarily applied to evaluate habitat suitability for a target species under specific environmental conditions and predict its potential habitat distribution within the study area. The model requires two key inputs: the current spatial distribution points of the target species and the environmental factors influencing its distribution [21]. By maximizing entropy, the MaxEnt model generates the most uniform probability distribution constrained by the provided data [22]. Essential inputs for the model include the geographic coordinates (latitude and longitude) of the target species’ occurrences and environmental variables, such as temperature, precipitation, and topographic features, that shape its distribution.

In this study, distribution data for the dhole (*Cuon alpinus*) in China were compiled from multiple sources including the *Red List of China’s Biodiversity* [23], the Global Biodiversity Information Facility (GBIF, https://www.gbif.org/, accessed on 29 August 2024), and the National Animal Collection Resource Center of China (http://museum.ioz.ac.cn/, accessed on 28 August 2024). A total of 232 occurrence records for the dhole were initially collected (Figure 1A). To mitigate the effects of spatial autocorrelation due to closely spaced points, geographic information system (GIS) tools, specifically ArcGIS 10.8, were used to filter out records with inter-point distances of less than 1 km [24,25]. After this process, 219 occurrence points remained and were subsequently used to develop the ecological niche model for the dhole. The boundaries of China and its provinces were sourced from the Resource and Environment Science and Data Center, Chinese Academy of Sciences (http://www.resdc.cn/, accessed on 28 August 2024) and imported into ArcGIS (version 10.8) software using the WGS84 coordinate system. After incorporating the dhole occurrence coordinates, a distribution map was generated to illustrate the current status of dhole populations across various provinces in China (Figure 1A).

### 2.2. Collection and Processing of Environmental Variables

We utilized 68 environmental factors related to dhole distribution as variables for model evaluation, primarily sourced from the global meteorological database (WorldClim version 1.4, http://www.worldclim.org/, accessed on 28 August 2024). These variables included elevation (alt), 19 bioclimatic factors (Bio1-Bio19), monthly precipitation (Prec1-Prec12), monthly average minimum temperature (Tmin1-12), monthly average maximum temperature (Tmax1-12), and monthly average temperature (Tmean1-12). Due to issues like autocorrelation and multicollinearity among environmental variables, which can impact model predictions [26], we addressed this by employing correlation coefficients for variable selection. Using SPSS 22.0 software, we extracted environmental attribute values corresponding to dhole distribution sample points and computed the correlation coefficients among variables. During the variable selection process, we specifically focused on highly correlated environmental factors (|r| ≥ 0.80) and systematically removed them based on the following considerations. Firstly, highly correlated environmental variables can introduce multicollinearity into the model. This issue leads to unstable parameter estimates and inflated standard errors, thereby diminishing the model’s explanatory power and predictive accuracy [27]. Secondly, retaining a large number of highly correlated variables increases the model complexity, which may result in overfitting. Overfitted models perform well on training datasets but poorly on new, unseen data [28]. Additionally, from a biological perspective, highly correlated variables often reflect the same ecological processes or environmental characteristics. Thus, removing one of these variables does not result in a loss of critical ecological information. Based on these considerations, we removed environmental factors with high correlation (|r| ≥ 0.80) and retained variables with lower correlation (|r| < 0.80) that were biologically meaningful. These selected variables were then incorporated into the MaxEnt model for computation, thereby enhancing the accuracy of the simulation results [27,28].

### 2.3. Optimization and Construction of the MaxEnt Model Parameters

This study employed the MaxEnt 3.3.3k model, incorporating the filtered occurrence data of dholes in China along with environmental variable datasets. For feature class selection, a combination of linear (L), quadratic (Q), product (P), threshold (T), and hinge (H) was applied, tailored to the characteristics of the species’ distribution data. Additionally, the “Regularization multiplier” (β) values were set within a range of 1.0 to 5.0, incrementing by 0.5, to generate AICc and BIC scores under varying parameter settings. The optimal parameters for constructing the habitat suitability model were determined based on the lowest AICc and BIC scores, obtained through ENMTools, while also considering the smoothness of the response curves.

To determine the optimal parameters, this study allocated 75% of the occurrence data for constructing the habitat suitability model for dholes, while the remaining 25% was reserved as a test set to validate model accuracy [29,30,31]. The jackknife method was applied for 10 iterations to evaluate the contribution of the environmental variables. Model performance was assessed using the receiver operating characteristic (ROC) curve and the area under the curve (AUC) value. The AUC value, ranging from 0.5 to 1, serves as a standard for evaluating model prediction accuracy, with higher AUC values indicating a stronger correspondence between the model-predicted spatial distribution and the actual distribution of the species [32].

### 2.4. Analysis of Habitat Suitability for Dholes in China

The results of the MaxEnt modeling for dhole habitat suitability in China were formatted as “.asc” files and imported into ArcGIS (version 10.8) software. Simultaneously, the provincial boundary layer of China was imported to generate a probability distribution map for dholes in China. The results, ranging from 0 to 1, indicated higher species presence probabilities with larger values. The ArcGIS reclassification function was employed to categorize the result layer into four levels: high suitability habitat (probability ≥ 0.6), moderate suitability habitat (probability between 0.4 and 0.6), low suitability habitat (probability between 0.2 and 0.4), and unsuitable habitat (probability < 0.2) [33].

## 3. Results

### 3.1. The Distribution Sites of the Dhole (Cuon alpinus) in China

Based on the literature and data from the GBIF, a total of 219 occurrence points for the dhole had been documented across 25 provinces, autonomous regions, and municipalities in China. The distribution data indicated that Guizhou had the highest number of occurrence points (n = 23), followed by Sichuan (n = 21), Xinjiang (n = 21), Yunnan (n = 18), and Gansu (n = 17). This information serves as a basis for constructing MaxEnt models for the species.

### 3.2. Environmental Variable Selection and Model Validation

Based on the correlation analysis, environmental variables with a correlation coefficient exceeding 0.8 were excluded from the model. Variables with lower correlations were retained for constructing the MaxEnt model for dholes in China, leading to the retention of eight variables (Table 1). These excluded variables comprised of factors related to topography, temperature, and precipitation. Given its significant informational value, the annual mean temperature (Bio1) was retained for subsequent analysis of the environmental factors influencing dhole habitat suitability.

Various parameter combinations were evaluated, and based on the calculated AICc and BIC scores, along with the smoothness of the fitted response curves, the optimal parameter combination was determined to be L + Q (linear, quadratic) with a β value of 1.5. The ROC curve generated by the MaxEnt model showed that the average AUC value from 10 repeated runs was 0.813, with a standard deviation (SD) of 0.034 (Figure 1B). These results indicated a high level of predictive accuracy, demonstrating the model’s robustness for further analyses.

### 3.3. Analysis of Environmental Factors Influencing Dhole Distribution

Among the eight selected environmental variables, the mean diurnal range (Bio2), temperature seasonality (Bio4), altitude, and minimum temperature of the coldest month (Bio6) demonstrated relatively high contributions to the spatial suitability model for dholes in China, with each exceeding 10% and cumulatively accounting for over 88.5% of the total contribution (Table 2). The response curves derived from the MaxEnt model revealed that dholes exhibited the highest suitability for distribution in regions with a mean diurnal temperature range of 4–5 °C, with suitability decreasing as the mean diurnal temperature range increased (Figure 2A). The species was most suited to areas with temperature seasonality between 1000 and 4000, with suitability declining as temperature seasonality increased (Figure 2B). The relationship with altitude was parabolic, with the highest suitability observed at altitudes between 2000 and 3000 m (Figure 2C). The minimum temperature of the coldest month followed a semi-parabolic distribution, with suitability increasing with rising minimum temperatures, peaking at 10–20 °C (Figure 2D). Annual precipitation also showed a parabolic distribution, with dholes most suited to areas receiving 1700–2500 mm of precipitation annually (Figure 2E). Similarly, precipitation seasonality displayed a semi-parabolic trend, with the highest suitability found in regions with precipitation seasonality values between 20 and 60 (Figure 2F).

Furthermore, we selected nine provinces with a great number of distribution points of dholes for a comparative analysis of the environmental factors (Appendix A, Table A1). As illustrated in the figure, there were notable differences in the distribution altitudes of dholes across the various provinces. Xizang exhibited the highest distribution altitude, while Guangxi Province showed the lowest (Figure 2G). Dholes in Xinjiang experienced relatively lower annual precipitation, minimum temperature during the coldest month, and overall annual precipitation, whereas dholes in Guangxi and Yunnan Provinces had comparatively higher values for these variables (Figure 2H,K,L). Dholes in Xizang displayed the highest mean diurnal temperature range, whereas those in Guizhou had the lowest (Figure 2I). Additionally, dholes in Yunnan Province had the lowest temperature seasonality, while those in Xinjiang had the highest (Figure 2J).

### 3.4. Analysis of Suitable Habitat Distribution for Dholes in China and Major Provinces

Using the key environmental variables and effective dhole distribution data, we constructed a maximum entropy model to simulate habitat suitability. The results showed that the suitable habitats for dholes were mainly located in central-western and northwestern China, with some sporadic areas in the southeastern areas. Highly suitable habitats were concentrated in the central-western region (Figure 3A).

In this study, Guizhou Province had the highest percentage of suitable area for dholes (99.76%), followed by Sichuan (99.14%) and Guangxi (96.36%). Yunnan (90.95%), Hubei (89.31%), and Shaanxi (73.96%) had a relatively higher percentage of suitable area for dholes. Gansu (50.01%) and Xizang (27.94%) had a relatively lower percentage of suitable area for dholes, and Xinjiang had the lowest percentage of suitable area for dholes (8.25%) (Table 2).

In Guizhou Province, most areas were suitable for dhole distribution, with an unsuitable distribution in sporadic southern regions. Highly, moderately, and lowly suitable areas for dhole distribution were 90,270 km^2^, 73,331 km^2^, and 12,139 km^2^, respectively, accounting for 51.24%, 41.63%, and 6.89% of the total area of Guizhou. The unsuitable area was 427 km^2^, accounting for 0.24% (Figure 3B). Dholes were mainly suitable for distribution in scattered regions in the central and eastern parts of Sichuan. The statistical analysis indicates that in Sichuan, the areas that are highly suitable, moderately suitable, and lowly suitable areas for dhole distribution were 80,245 km^2^, 202,311 km^2^, and 199,272 km^2^, respectively, accounting for 16.51%, 41.63%, and 41.00% of the total area of Sichuan. The unsuitable area was 4172 km^2^, accounting for 0.86% (Figure 3C). In Guangxi Province, dholes were mainly suitable for distribution in scattered regions in the northeast, northwest, and southwest. Highly, moderately, and lowly suitable areas were 5.27 km^2^, 25.76 km^2^, and 65.33 km^2^, respectively, accounting for 2.50%, 21.53%, and 25.98% of the total area. The unsuitable area was 8640 km^2^, accounting for 3.64% (Figure 3D; Table 2).

In Yunnan Province, most areas were suitable for dhole distribution, with unsuitable areas in sporadic regions in the western, northeastern, and southeastern areas. Highly suitable, moderately suitable, and lowly suitable areas for dhole distribution, covering 30,912 km^2^, 162,594 km^2^, and 164,912 km^2^, respectively, accounted for 7.84%, 41.26%, and 41.85% of the total area of Yunnan. The unsuitable area was 35,682 km^2^, accounting for 9.05% (Figure 3E). Dholes were mainly suitable for distribution in the western sporadic regions of Hubei. The statistical analysis indicated that in Hubei Province, areas highly suitable, moderately suitable, and lowly suitable for dhole distribution were 37,529 km^2^, 22,461 km^2^, and 106,032 km^2^, respectively, accounting for 20.19%, 12.08%, and 57.04% of the total area of Hubei. The unsuitable area was 19,878 km^2^, accounting for 10.69% (Figure 3F). Dholes were mainly suitable for distribution in the southern part of Shaanxi. The statistical analysis showed that in Shaanxi, the areas that were highly suitable, moderately suitable, and lowly suitable for dhole distribution were 28,050 km^2^, 34,003 km^2^, and 69,456 km^2^, respectively, accounting for 13.64%, 16.54%, and 33.78% of the total area of Shaanxi. The unsuitable area was 74,114 km^2^, accounting for 36.04% (Figure 3G).

Furthermore, dholes were mainly suitable for distribution in the southern and central regions of Gansu. Highly, moderately, and lowly suitable areas were 10,637 km^2^, 91,710 km^2^, and 110,653 km^2^, respectively, accounting for 2.50%, 21.53%, and 25.98% of the total area. The unsuitable area was 212,899 km^2^, accounting for 49.99% (Figure 3H). Dholes were primarily suitable for distribution in the southeast and western sporadic regions of Xizang. The statistical analysis revealed that Xizang had areas that were highly suitable, moderately suitable, and lowly suitable for dhole distribution, covering 22,121 km^2^, 68,321 km^2^, and 245,603 km^2^, respectively, accounting for 1.84%, 5.68%, and 20.42% of the total area of Xizang. The unsuitable area was 866,755 km^2^, accounting for 72.06% (Figure 3I). In Xinjiang Province, most areas were suitable for dhole distribution, with a suitable distribution in sporadic southwest and northwest regions. The statistical analysis showed that Xinjiang had areas highly suitable and moderately suitable for dhole distribution, covering 25,191 km^2^ and 112,118 km^2^, respectively, and accounting for 1.51% and 6.73% of the total area of Xinjiang. The unsuitable area was 1,527,591 km^2^, accounting for 91.75% (Figure 3J; Table 2).

## 4. Discussion

### 4.1. Key Environmental Factors Influencing Distribution of Dholes in China

In this study, the mean diurnal range, temperature seasonality, altitude, and minimum temperature of the coldest month showed relatively high relative contributions to constructing the habitat suitability of dholes in China, which indicates that temperature variation, extreme low temperature and elevation significantly influences the distribution of suitable habitats for dholes.

Temperature variation can directly affect the spatial distribution of wildlife. For instance, Royle’s pika (*Ochotona roylei*) is particularly vulnerable to rising temperatures, which hinder their surface activity and dispersal, reducing their potential distribution due to climate change [34]. On the contrary, natural suboptimal conditions caused by low temperatures for the bobak marmot (*Marmota bobak*) also reduce their surface activity and limit their distribution [35]. Additionally, temperature variation can indirectly affect wildlife by altering habitat characteristics and food resources [36]. Brown bears (*Ursus arctos*) have had to change their foraging strategies and activity ranges due to the reduced availability of food resources caused by global warming [37]. Snow leopards (*P. uncia*) have shifted their distributions in response to the climate-driven prey reduction in the Himalayas [38]. Dholes mainly prey on ungulates such as water deer (*Rusa unicolor*), muntjac (*Muntiacus*), and wild boar (*Sus scrofa*) [39,40]. Growing evidence suggests that most ungulates may migrate to higher altitudes in cooler regions in response to global warming [41,42]. This shift in ungulate distribution patterns is likely to impact the availability of dhole food resources, thereby affecting the distribution of suitable habitats for dholes in China.

Extreme low temperatures can influence the spatial distribution of dholes by impacting their prey resources. Extreme low temperatures can increase energy expenditure and lead to adverse weather conditions (e.g., Snowstorm), which likely constrain the movements and foraging rates of ungulates [43,44], thereby indirectly affecting the dhole distribution in China. Studies on the North American goat (*Oreamnos americanus*) have demonstrated that their daily/weekly movements decrease with increasing cumulative snowfall during winter [45]. Similarly, snow accumulation can reduce the encounter rates of red deer (*Cervus canadensis*) with vegetation and increase the time required for foraging [46]. Furthermore, the majority of ungulates are phytophagous [47,48], and extreme low temperatures can inhibit plant growth and development [49,50]. Consequently, a shortage of vegetation resources may alter the distribution patterns of ungulates, which in turn could influence the distribution of dholes.

Altitude is a critical factor influencing the distribution of suitable habitats for the dhole in China. The dhole population is closely tied to forest ecosystems, which are marked by substantial altitudinal gradients [51,52]. Furthermore, the primary prey species of the dhole, such as ungulates, also inhabit these forested areas [11,53]. Altitudinal variations can affect the habitat selection and distribution of these prey species [54,55], which in turn may influence the distribution of potentially suitable habitats for dholes in China.

### 4.2. Environmental Factors Influencing Distribution of Dholes in Different Provinces

The environmental factors influencing dhole distribution varied across the nine provinces in China, reflecting the specific suitability of each region for the species. Our results showed that Xizang and Guangxi had the highest and lowest areas with altitudes suitable for dholes. These altitudinal differences are intricately linked to China’s topography, which is characterized by high western regions and low eastern regions. The eastern part of China is primarily composed of plains and hills such as the Huabei Plain and Shandong Hills, whereas the western part is dominated by plateaus including the Qinghai-Xizang Plateau (https://www.gov.cn/, accessed on 26 August 2024). The variations in temperature and annual precipitation across provinces are likely due to their distinct climatic types. For example, the lowest temperature seasonality was found in the dhole distribution in Yunnan, while the highest was noted in Xinjiang. Yunnan is classified as belonging to the subtropical plateau monsoon climate zone. The climate is typified by a prevalence of heat, the simultaneous occurrence of precipitation and heat, and the presence of distinct four-seasonal cycles. Additionally, the seasonal temperature variation is not pronounced [50]. However, Xinjiang is classified as belonging to the temperate continental climate. This climate type is typified by dry and low precipitation levels due to its distance from the ocean and the presence of mountainous terrain. Seasonal temperature fluctuations are also more pronounced compared with other climate types [56].

### 4.3. Distribution Pattern of the Potential Suitable Habitat for Dholes in China

In this study, the suitable distribution areas for dholes were mainly located in the central-western, northwestern, and sporadic regions of the southeastern part of China. Highly suitable distribution areas were primarily concentrated in the central-western regions. The central-western regions of China are characterized by a higher concentration of mountains and forest cover, which collectively contribute to a rich diversity of wildlife resources [57,58]. The presence of adequate prey resources may be a significant contributing factor to the high suitability of the area for dholes. Furthermore, the human population density of the central-western regions, particularly the western region, is comparatively lower than that of the eastern and southern regions of China [59]. Numerous studies have demonstrated that human disturbance has a profound impact on the survival and distribution of wild animals [60,61]. Research on the sympatric forest antelope has shown that human activity negatively affects the temporal activity of *Philantomba monticola* as well as the spatial distribution of *Tragelaphus scriptus* and *Sylvicapra grimmia* [62]. Therefore, the lower level of human disturbance in the central-western regions is likely another significant factor contributing to the high suitability for dholes.

### 4.4. Distribution Pattern of the Potential Suitable Habitat for Dholes in Different Provinces

Among the nine major provinces in China where dholes are distributed, Guizhou, Sichuan, and Guangxi had the highest percentages of suitable habitat area. These provinces play a crucial role in China’s biodiversity conservation efforts. Of the 32 priority areas (inland and water) for biodiversity conservation in China, nine priority areas cover parts of the Guizhou, Sichuan or Guangxi Provinces including the south section of Hengduan Mountain, Min Mountains–north section of Hengduan Mountain, and Wuling Mountain biodiversity reserve priority areas (Scope of priority areas for biodiversity conservation in China, 2015; https://www.mee.gov.cn/, accessed on 26 August 2024). Furthermore, the central and western regions of Sichuan are located within the Mountains of southwest China, which are recognized as one of the 34 global biodiversity hotspots (Critical Ecosystem Partnership Fund Ecosystem Profile: Mountains of Southwest China Hotspot, 2002 (https://www.cepf.net/, accessed on 27 August 2024). More attention to these three provinces is needed for the further conservation and management of dholes in China.

### 4.5. Research Limitations

The limitations of this study are twofold. Firstly, the data sources were primarily derived from the literature and the GBIF platform, which did not include data from infrared camera traps. This limitation may restrict the comprehensiveness of our results. Secondly, although integrating the national and provincial population density data of dholes would enhance the robustness of our findings, such comprehensive reports are currently unavailable in China. More diverse data sources including infrared camera traps will be collected in our further research to provide a more complete picture of dhole populations.

## 5. Conclusions

In this study, we employed the MaxEnt model to assess the habitat suitability of the dhole in China. The results demonstrated the following. (1) The key factors impacting the distribution of potential suitable habitats for the dhole in China are temperature changes (mean diurnal range of 4–5 °C and temperature seasonality between 1000 and 4000), extreme temperature (minimum temperature of the coldest month between 10 and 20 °C), and elevation (2000–3000 m). (2) Potential suitable habitats for dholes are mainly located in the central-western, northwestern, and sporadic regions of the southeastern part of China. The areas identified as highly suitable for the species were primarily concentrated in the central-western region. (3) The proportion of suitable habitats for dholes differed significantly between the nine provinces in China. Guizhou, Sichuan, and Guangxi had the highest percentage of area with suitability for dholes in China. This study revealed the distribution patterns of suitable habitats and the dominant environmental factors limiting the distribution of dholes in China. The findings will provide ecological data support for the conservation and management of dholes in China.

## Figures and Tables

**Figure 1 animals-15-00463-f001:**
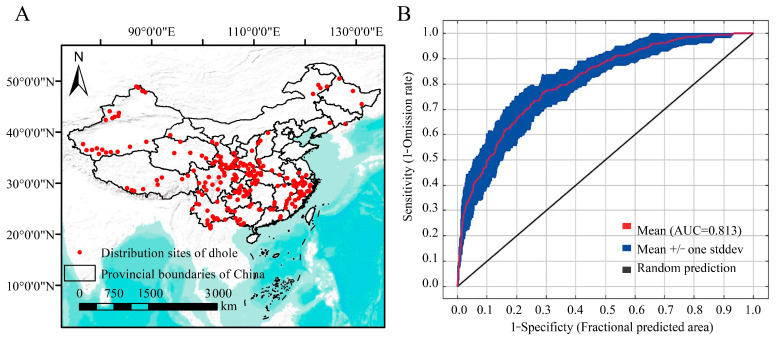
Map of dhole distribution points in China (**A**) and the accuracy analysis curve of the MaxEnt simulated dhole spatial distribution (**B**).

**Figure 2 animals-15-00463-f002:**
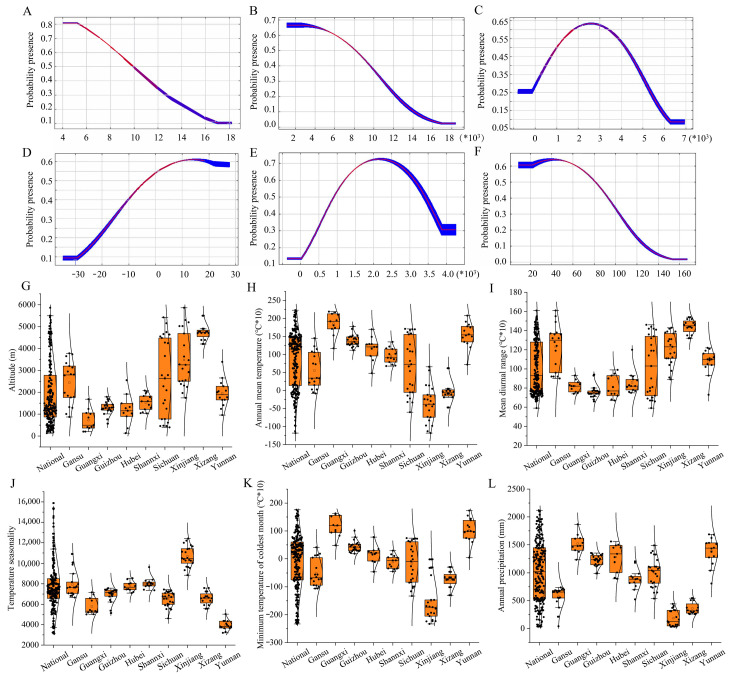
Analysis of environmental factors influencing dhole distribution including mean diurnal range (**A**), temperature seasonality (**B**), altitude (**C**), minimum temperature of coldest month (**D**), annual precipitation (**E**), and precipitation seasonality (**F**). Comparison and analysis of the environmental factors across different provinces including altitude (**G**), annual mean temperature (**H**), mean diurnal range (**I**), temperature seasonality (**J**), minimum temperature of coldest month (**K**), and annual precipitation (**L**).

**Figure 3 animals-15-00463-f003:**
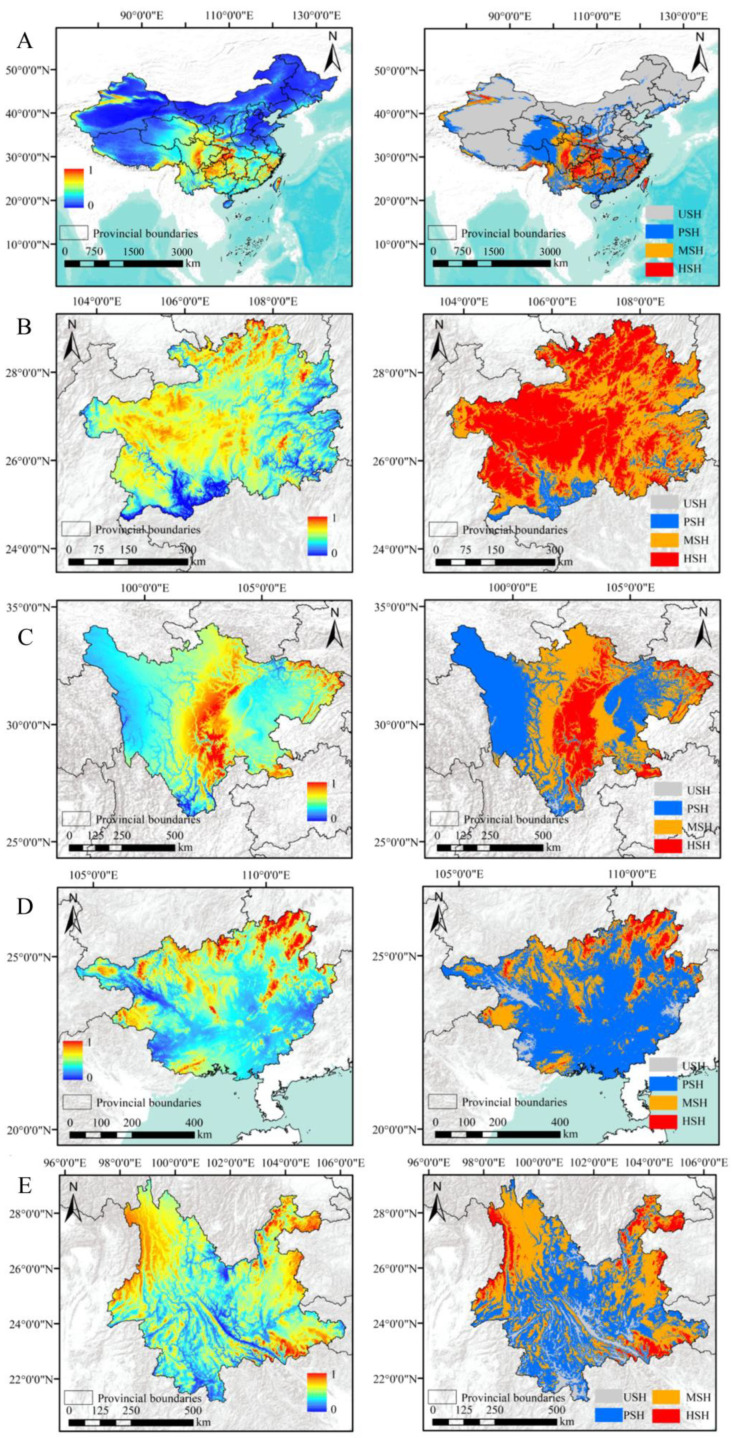
Habitat suitability distribution analysis of dholes in China (**A**) and the major provinces including Guizhou (**B**), Sichuan (**C**), Guangxi (**D**), Yunnan (**E**), Hubei (**F**), Shaanxi (**G**), Gansu (**H**), Xizang (**I**), and Xinjiang **(J**).

**Table 1 animals-15-00463-t001:** Analysis of the relative contribution of environmental factors in constructing the MaxEnt model for dholes in China.

Variables	Percent Contribution (%)
Mean diurnal range (Bio2, °C)	38.7
Temperature seasonality (Bio4)	20.2
Altitude (Alt, m)	19.2
Minimum temperature of coldest month (Bio6, °C)	10.4
Annual precipitation (Bio12, mm)	7.7
Precipitation seasonality (Bio15)	2.3
Isothermality (Bio3)	0.8
Monthly precipitation in November (Prec11, mm)	0.7

**Table 2 animals-15-00463-t002:** Habitat composition analysis of dholes in different provinces of China.

	Province	HSH	MSH	PSH	USH	SUM
Area (km^2^)	Guizhou	90,270	73,331	12,139	427	176,167
	Sichuan	80,245	202,311	199,272	4172	486,000
	Guangxi	12,523	61,205	155,232	8640	237,600
	Yunnan	30,912	162,594	164,912	35,682	394,100
	Hubei	37,529	22,461	106,032	19,878	185,900
	Shannxi	28,050	34,003	69,456	74,114	205,624
	Gansu	10,637	91,710	110,653	212,899	425,900
	Xizang	22,121	68,321	245,603	866,755	1,202,800
	Xinjiang	25,191	112,118	0	1,527,591	1,664,900
Proportion (%)	Guizhou	51.24	41.63	6.89	0.24	100
	Sichuan	16.51	41.63	41.00	0.86	100
	Guangxi	5.27	25.76	65.33	3.64	100
	Yunnan	7.84	41.26	41.85	9.05	100
	Hubei	20.19	12.08	57.04	10.69	100
	Shannxi	13.64	16.54	33.78	36.04	100
	Gansu	2.50	21.53	25.98	49.99	100
	Xizang	1.84	5.68	20.42	72.06	100
	Xinjiang	1.51	6.73	0.00	91.75	100

Abbreviations: HSH, highly suitable habitat; MSH, moderately suitable habitat; PSH, partially suitable habitat; USH, unsuitable habitat.

## Data Availability

The original contributions presented in this study are included in the article. Further inquiries can be directed to the corresponding author.

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
