# Peer review of "Distribution Patterns and Ecological Determinants of Suitable Habitats for the Dhole (Cuon alpinus) in China"

_animals, 2025, doi:10.3390/ani15040463_

Round 1

Reviewer 1 Report

Comments and Suggestions for Authors

The manuscript reports an interesting desktop study that aimed to model (using MaxEnt) habitat suitability for the Dhole in China, and mainland provinces therein. The authors made some effort to gather geographic locations of Dhole sightings from various databases. These locations were refined by imposing a distance of >1km (line 105) to avoid autocorrelation. This distance seems arbitrary and the citation of a study on musk deer of questionable relevance. The Dhole is highly social and lives in clan groups. Canids of its size typically have quite large home ranges. Thus, the authors need a better justification for  inclusion/exclusion of sightings. The authors gather geographical and climate data from authoritative sources to use in the MaxEnt model. They address autocorrelation and collinearity and refine an initial set of 68 variables down to 8. Confusingly (line 158-9), they write “leading to the removal of eight variables”. I presume they meant the ‘retention of eight variables’ since these variables are referred to in relation to habitat suitability. Of the Dhole sightings, they state that “25% was reserved as a test set to validate model accuracy” (line 135-6). It is unclear how this set was used and how the 25% were selected (random draw?). The authors present Fig. 1B as evidence that their model is significantly better than random. It would be more convincing to include a figure of the distribution of their test set on their map of habitat suitability (e.g. Fig 3A). In section 3.3 (line 214-315), the authors confuse the agent of suitability which is the habitat not the Dhole. For instance, “dholes were mainly suitable for distribution in most areas” (line 230). What is presumably meant is ‘most areas were suitable for Dholes’ (according to their model). In section 4.3 (line 337-353), the authors introduce other variables like “wildlife resources” and “human disturbance”. Are these qualifying variables to the habitat suitability model or should they have been in the model?

I note a few minor issues that should be addressed as follows:

Line 33: provide critical

Line 49: access date for URL, here and elsewhere throughout the text

Line 275: The reader has to guess what the column headings HSH, MSH, PSH and USH are; presumably, high, medium, partial and un suitable habitat. The Table heading should explain these.

Line 281: bit of confusion about intention of model – spatial suitability or habitat suitability?

Line 299: (e.g. snowstorm)

Line 318: It is the suitability of environmental factors for the Dhole not the differences per se.

Line 320: lowest areas with altitudes suitable for dholes.

Line 366: more attention to these three provinces is needed for

Reviewer 2 Report

Comments and Suggestions for Authors

Dear Authors,

The manuscript certainly touches upon an important topic. The study touches upon a predator from the Canidae family. The study of rare and declining animals is always relevant. At the same time, it is necessary to improve the manuscript. Add some additions and clarity to it. In the Introduction, write to what type of animal the dhole belongs according to its daily activity. In the methodological aspects, it is necessary to add how this species is counted. Accept after minor revisions (add the dhole population density for China and its individual provinces). These data will make the authors' results in the manuscript more convincing.

Reviewer 3 Report

Comments and Suggestions for Authors

The authors present a study on the Dhole in China. This study explores the distribution patterns of suitable habitats for this species and highlights the key ecological factors influencing their habitat suitability. The analyses indicate that temperature variation, extremely low temperatures, and elevation significantly shape these habitats. China's central-western and northwestern regions are identified as primary areas of suitable habitat, with notable concentrations in Guizhou, Sichuan, and Guangxi provinces.

The study is well-planned, the analyses are appropriate, and the manuscript is well-written. However, in the methods and results sections, you need to explain the reasoning behind excluding highly correlated variables and provide a more detailed explanation of why specific variables were retained or excluded. Also, the authors mention using 75% of the occurrence data for model construction and 25% for testing. Implementing k-fold cross-validation could provide a more comprehensive assessment of model performance and reduce the risk of overfitting.

In the discussion, a more thorough evaluation of the study's limitations, such as potential biases in occurrence data sources or the generalization of the model results, would provide a balanced view of the findings and their applicability to conservation efforts.

Line 68 – I do not wish to get into this political embroilment, whether Taiwan is or is not a province of China. I recommend sticking to mainland China without bringing territorial disputes into the scientific literature. This is especially true since you relate to the other provinces but not Taiwan in your analyses and Discussion. I will let the editors decide how to handle this one.

Round 2

Reviewer 1 Report

Comments and Suggestions for Authors

The authors have adequately addressed my comments and suggestions in their revised manuscript. I thank them for clarification of training/testing datasets.